# The Status of p53 Oligomeric and Aggregation States in Cancer

**DOI:** 10.3390/biom10040548

**Published:** 2020-04-04

**Authors:** Guilherme A. P. de Oliveira, Elaine C. Petronilho, Murilo M. Pedrote, Mayra A. Marques, Tuane C. R. G. Vieira, Elio A. Cino, Jerson L. Silva

**Affiliations:** 1Institute of Medical Biochemistry Leopoldo de Meis, National Institute of Science and Technology for Structural Biology and Bioimaging, National Center of Nuclear Magnetic Resonance Jiri Jonas, Federal University of Rio de Janeiro, Rio de Janeiro RJ 21941-901, Brazil; mrguioli@gmail.com (G.A.P.d.O.); nane.cp@gmail.com (E.C.P.); murilopedrote@gmail.com (M.M.P.); mayra.marques@ymail.com (M.A.M.); tuane@bioqmed.ufrj.br (T.C.R.G.V.); 2Department of Biochemistry and Immunology, Federal University of Minas Gerais, Belo Horizonte MG 31270-901, Brazil

**Keywords:** p53 aggregation, p53 oligomers, gain-of-function effects, mutant p53, oncogenesis

## Abstract

Despite being referred to as the guardian of the genome, when impacted by mutations, p53 can lose its protective functions and become a renegade. The malignant transformation of p53 occurs on multiple levels, such as altered DNA binding properties, acquisition of novel cellular partners, or associating into different oligomeric states. The consequences of these transformations can be catastrophic. Ongoing studies have implicated different oligomeric p53 species as having a central role in cancer biology; however, the correlation between p53 oligomerization status and oncogenic activities in cancer progression remains an open conundrum. In this review, we summarize the roles of different p53 oligomeric states in cancer and discuss potential research directions for overcoming aberrant p53 function associated with them. We address how misfolding and prion-like amyloid aggregation of p53 seem to play a crucial role in cancer development. The misfolded and aggregated states of mutant p53 are prospective targets for the development of novel therapeutic strategies against tumoral diseases.

## 1. Overview of Mutated p53 Domains in Cancer

p53 was identified by David Lane and Arnold Levine in extracts of SV40-transformed mouse cell lines [1,2] and has become a prominent topic in cancer biology. The transcription factor p53 has tumor suppressor activities when cells are threatened by exogenous mutagens, telomere erosion, and hypoxia [3] but also presents antagonistic phenotypes [4,5]. As a tumor suppressor, it regulates cell cycle arrest, senescence, and apoptosis [6,7]. These are common p53 actions to avert oncogenic transformations and tumor occurrence. In resting cells, p53 transcriptional activity is tightly controlled by the E3 ubiquitin protein ligase, MDM2, leading to p53 degradation and rapid turnover [8]. When active, p53 tetramers bind to responsive elements (RE), and regulate a variety of genes involved in tumor suppressor activities [6,7,9].

*TP53* is one of the most frequently mutated genes in human cancer, with most alterations occurring in the region coding for the DNA binding domain (DBD) [10]. p53 is composed of three domains, an N-terminal transactivation domain (TAD, residues 1–70), DBD (residues 94–293), and an oligomerization domain (OD, 324–355) (Figure 1). The TAD has intrinsic flexibility, and the DBD is flanked by two disordered regions containing proline-rich motifs (PRM, residues 71–93 and 294–323). The C-terminus region consists of an unstructured basic segment (residues 356–393). These flexible segments are of pivotal importance triggering molecular recognition and regulating p53 transcriptional activity. The so-called “molecular antennas” are mostly tuned by posttranslational modifications such as phosphorylation and acetylation [11,12]. For example, DNA damage leads to a cascade of TAD phosphorylation, MDM2 dissociation, and p300/CTB binding, which ultimately favors p53 transcriptional activity [13,14]. The first p53 PRM mediates p53-dependent apoptosis but is dispensable for cell growth arrest [15]. C-terminal acetylation may also have a role in regulating DNA binding [16,17].

There are a substantial number of somatic mutations reported within the DBD which can lead to specific phenotypes. Among them, six sites (i.e., R175, G245, R248, R249, R273, and R282) are classified as hotspots [18] (Figure 1). Mutations at sites R248, R273, and R280 disrupt DNA binding and transcriptional activity, and are known as contact mutations. Other sites, such as R175, G245, and R249 are important for the structural stability of p53. Mutations within p53 PRMs and OD are not common, but there are some exceptions. For instance, there are cases of germ-line substitutions at PRM position 82, leading to a Pro-to-Leu substitution, and somatic mutations in bladder tumors at position 85 and 89 resulting in Pro-to-Ser substitutions [19,20]. The *TP53* germ-line mutations predispose subjects to a variety of tumor types. This autosomal-dominant cancer predisposition is also known as Li–Fraumeni syndrome. One particular case occurs within the OD at position 337, leading to an Arg-to-His substitution. This is the most frequent p53 germ-line mutation found to date [18,21], and is almost exclusively identified in families in southern Brazil.

p53 mutations can impact its functionality in different ways. They may lead to a loss-of-function phenotype, where p53s’ ability to bind DNA is compromised [22]. Second, they may alter p53s’ conformation, leading to gain-of-function (GoF), or oncogenic activity [4]. In this situation, the novel conformation might have the ability to form different oligomeric states, gain or lose affinity to responsive elements, transcription factors, or bind to other regulatory proteins [5,23]. p53 mutations can also influence the activity of wild-type p53 by dominant-negative regulation [24].

## 2. Renegade p53 Outcomes

p53 mutations can cause it to lose its protective roles and obtain tumor-promoting functions. The molecular mechanisms underlying this transformation remain elusive, though it has been reported that p53 GoF phenotypes include increased migration, invasiveness, angiogenesis, stem cell expansion, survival, proliferation, tissue remodeling, chemoresistance, genomic instability, and others (Reference [4] and citations therein). The broad range of phenotypes reflects the multimodal participation of p53 in signaling pathways. There are a few scenarios which may explain how mutated p53 could contribute to malignancy: (i) conformational changes in mutated p53 may affect its modus operandi for binding DNA; (ii) mutation may affect p53’s interactome, leading it to bind other transcription factors, accessory proteins, and kinases; (iii) mutations may shift the conformation distribution to an ensemble favoring different types of aggregated species, including those presenting amyloid features (Figure 2).

P53 mutations do not necessarily abolish its DNA-binding capabilities, but in some GoF p53 mutations, substitution completely abrogates the ability to bind DNA. Mutations may change the modus operandi of binding, for instance, increasing or decreasing its affinity to DNA, acquiring sequence-specific preferences, or modifying the chromatin packing (Figure 2). For example, mutant p53 has been shown to induce histone acetylation through recruitment of CBP and STAT2 on promoter regions impacting the expression of genes that participate in cell proliferation, motility, and tumorigenicity [25]. Additionally, mutant p53 can bind the transcription factor (TF) ETS2, which interacts with MLL and MOZ histone methyl- and acetyl-transferases [26]. Moreover, certain mutants can bind to matrix attachment DNA sequences with higher affinity than wild-type p53 [27]. Mutated p53 has a large repertoire of DNA-binding sequences as reported by ChIP-on-chip analysis of p53 R175H extracted from breast cancer cells [28]. The results show that mutant p53 binds to 40 out of 154 analyzed promoters, and it influences the transcriptional output of NF-kB target genes [28].

The presence of cancer-related mutations also impacts p53 binding to other TFs, controlling expression of their target genes either positively or negatively. For instance, topoisomerase binding protein 1 (TopBP1) binds to mutant p53 and promotes p300 recruitment to NF-Y target gene promoters. It has been shown that TopBP facilitates mutant p53 interaction and inhibition of p63 and p73 transcriptional activities [29]. Furthermore, mutant p53 has the ability to prevent p63 and p73 DNA-binding activity [29,30], and reposition p63 to other DNA locations [31]. There is growing evidence of p53-induced TF modulation when cancer-related p53 mutations are present, reinforcing its oncogenic transformation in cancer [4]. Another novel action of mutant p53 is constitutive interaction with phosphatidylinositol phosphate kinase (PIPKI) and its product phosphatidylinositol 4,5-biphosphate [32]. This interaction recruits small heat shock proteins (sHSPs) to stabilize mutant p53 in the nucleus [32]. Rather than exclusively impacting transcriptional machineries, there are cases in which mutant p53 affects proteins unrelated to transcriptional activity such as nucleases, isomerases, and cell-cycle regulators. Interaction of mutant p53 with the nuclease Mre11 abrogates its association at double-stranded break sites, ultimately impairing homologous recombination [33]. Another example is the binding of mutant p53 to topoisomerase 1 (Top1) which potentially increases mutagenic DNA rearrangements [34]. 

The last scenario hypothesizes that mutations may change the oligomerization status of p53, leading to a variety of oligomerization-dependent functionalities. p53 can assume a variety of states, from active tetramers and octamers, to amyloid species, fibrils, and amorphous aggregates. Tetramers and octamers of p53 bind to DNA, but the oligomeric state of cytosolic p53 and its function is the subject of debate. On the other hand, the topic of p53 aggregation is more complicated and less understood. These processes occur via complex mechanisms which are discussed in the subsequent sections, along with the roles of other molecules such as RNA and glycosaminoglycans.

## 3. p53 Oligomeric States

Different p53 functions often correlate with distinct oligomeric states. p53 dimerization is a normal molecular association event that occurs with a K_d_ of ~1 nM by association of the OD with a second monomer through an antiparallel β-sheet [35]. To form tetramers, dimers associate through a distinct helix–helix interface with a K_d_ of 100–1000 nM [35,36] (Figure 1 and Figure 3A). While tetramers are believed be the principal transcriptionally active species, larger assemblies can also bind DNA. Seminal studies have explored the quaternary composition of p53 in vitro. Using non-denaturing gradient electrophoresis, it has been shown that p53 migrates as tetramers and multiples of tetramers [37]. Further investigations revealed the ability of single p53 tetramers to bind separate DNA half-sites [38]. Electron microscopy (EM) and scanning transmission EM identified tetrameric p53 as the dominant DNA binding form; however, 12- and 16-mer oligomers were identified to participate in DNA looping events [39]. Tetramers were visualized to align along the DNA both in tandem, and perpendicular (stacked), the latter arising from quaternary interactions among the tetramers. The authors also discovered that p53 has a non-tetrameric OD between residues 1 and 320 that provides an additional interface for protein–protein interactions [39]. Crystallographic studies have shown that one p53 tetramer binds to one RE composed of two decametric palindromic half-site sequences. However, cryo-EM studies revealed that full-length p53 tetramers potentially bind to only one half-site of the RE [40], and later studies revealed that two tetramers interact specifically with one DNA RE at the same time [41]. Additional p53-DNA arrangements have also been identified, including a fully specific complex of two p53 dimers bound to two specific half-sites, and a hemispecific complex of one dimer bound to a specific binding site, and the second to an adjacent spacer sequence [42] (Figure 3A). Fluorescence correlation spectroscopy (FCS) showed that under resting conditions, a chemoresistant p53 mutant showed a higher population of cytosolic tetramers, compared to wild-type p53 [43]. Gaglia et al. [44] quantified the fraction of p53 monomers, dimers, and tetramers in living cells under resting and post-DNA damage conditions. They found that under resting conditions, p53 is present in a mixture of oligomeric states with a large cell-to-cell variability. After DNA damage, p53 tetramers rapidly form and accumulate [44]. In cell-based assays, while transiently over expressed, wt and p53 DNA-contact mutations revealed monomers, tetramers, and octamers on Western blots, aggregating hotspot mutations, including R175H and R249S, revealed larger multimeric assemblies [45]. Overall, these demonstrations illustrate that multiple oligomeric states are able to assemble and potentially bind to DNA, regulating gene expression. The quaternary organization of p53 can also affect its non-transcriptional functions within the cytosol [46]. Because the p53 tetramerization domain displays a leucine-rich nuclear export signal (NES) that is occluded in active tetramers, but not in dimers, cytosolic p53 is presumably dimeric [47]. In support of that, MDM2 exhibits superior binding to dimers than tetramers [48], reinforcing the scenario of a greater content of p53 dimers in the cytosol. Cytosolic p53 is involved in induction of mitochondrial outer membrane permeabilization (MOMP) [49] which ultimately results in the release of pro-apoptotic effectors and inhibition of autophagy (Reference [46] and citations therein) (Figure 3B). Further studies are needed to understand the transcriptional and non-transcriptional roles of different p53 assemblies, possible alterations in cancer cells, and their consequences.

Although oligomerization of p53 occurs primarily through its OD, higher order aggregate structures have been identified to nucleate via noncanonical segments [39]. Mutations within its DBD can make p53 more susceptible to form oligomeric species with amyloid-like features and oncogenic activities [50]. Aggregated species of mutant p53, and different p53 isoforms have been reported in cancer cells presenting GoF phenotypes [43,50,51,52,53]; however, their correlation with oncogenic activities in cancer requires clarification [4]. The Δ40p53 isoform in endometrial carcinoma has shown aggregation tendencies [54]. Mutant p53 oligomeric species larger than its tetrameric form have been detected in the nuclei in vivo, as well as evidence of tetramers in the cytosol [43]. The particular p53 mutation examined in the study is highly expressed, and accumulates as amyloid oligomers in glioblastoma cells presenting a chemoresistant phenotype [43]. It is plausible that some mutant p53 quaternary compositions, such as those depicted in Figure 3, could be involved in targeting of p53 to novel REs involved with chemoresistance genes. Still, the structural composition of these active amyloid oligomers remains enigmatic, as most have been identified by cell-based strategies. 

Our group pioneered, 17 years ago, the discovery that p53 aggregation would be related to both the dominant negative effect and the gain in p53 mutation function in cancer [55]. The first two papers pointing to the amyloid character of the p53 aggregation and their relevance to cancer were published in 2003 [55,56]. In 2012. we reported that p53 has typical amyloid features, and the propensity to seed aggregation in vitro. Seeds of the R248Q p53 mutant were able to accelerate wild-type p53 aggregation [50]. Further, mutant p53 co-localizes with amyloid-like protein aggregates in several breast cancer tissues [57] (Figure 3C). Other studies have shown the relationship between p53 amyloid aggregation and GoF in vivo. Xu et al. [45] discovered that mutations destabilizing the p53 DBD trigger coaggregation with wild-type p53 and its paralogs p63 and p73 (Figure 3C). They also found that p53 aggregation leads to increased expression of heat-shock protein Hsp70, an antiapoptotic factor [45]. Yang-Hartwich et al. [53] found an association between p53 aggregation and platinum chemoresistance in high-grade serous ovarian carcinoma. They observed that p14ARF inhibits the MDM2-mediated p53 degradation, culminating in p53 aggregation (Figure 3C). Another recent study demonstrated that the GoF acquired by mutated p53 (R248L) in glioblastoma promotes inflammation through positive regulation of the chemokine ligand C-C 2 (CCL2) and the tumor necrosis factor alpha (TNF-α), via signaling of the nuclear factor kappa B (NFkB) [58]. This gain of oncogenic function promotes increased microglia and inflammation in the central nervous system, accelerating glioblastoma progression and treatment resistance [58]. Similarly, amyloid aggregates of mutant p53 have been linked to temozolomide chemoresistance in aggressive glioblastoma [43]. Cancer cells presenting a chemoresistance p53 mutation display a GoF mechanism by over-expressing MGMT, an O^6^-methylguanine DNA-methyltransferase, that repairs DNA damage caused by temozolomide [43,59] (Figure 3C). By means of immunofluorescence, we discovered that the M237I p53 mutation colocalizes with amyloid oligomers in these glioblastoma cells [43]. Further, we observed by fluorescence correlation spectroscopy that this chemoresistant p53 mutation populates oligomers larger than tetramers within the nuclei of living cells [43]. This finding links the role of oligomers as transcriptionally active species. It is possible that some oligomeric or aggregated p53 species act as protagonists to GoF activities, such as chemoresistance. This might occur by direct binding to novel DNA regulatory elements, impacting the expression of genes related to the oncogenic phenotype, binding to other nuclear or cytosolic partners, or by other non-transcriptional activities of aggregated p53 

## 4. Other Factors Contributing to p53 Aggregation

The intrinsically disordered profile of p53 makes it “sticky” and enables association with unusual partners. Besides DNA, p53 binds to various other molecules, and such interactions can modulate its aggregation. p53 can also interact with RNA and RNA binding proteins. 

One typical stress response is the recruitment of nuclear proteins to the nucleolus (a membrane-less nuclear compartment that participates in the assembly of ribosome). This process involves the recruitment of proteins, such as p53 and MDM2, that accumulate as dense aggregates with polyadenylated RNA [60,61]. It is unclear how these structures are formed, but one hypothesis is that the process involves a phase separation [61]. Although phase separation is crucial in several cellular processes, the accumulation of proteins and nucleic acid in condensates may proceed through a liquid-solid phase transition that can lead to amyloidogenesis [61,62]. Nucleolus phase separation and amyloid formation can be promoted/seeded by RNA [63,64,65,66]. Recently, it was proposed that the liquid-liquid phase separation (LLPS) of p53 might be related with its cellular functions [65]. Spherical droplets can be formed in vitro, using a thermostable p53 mutant, at neutral and slightly acidic pH and at low salt concentrations. The N- and C-terminal disordered domains (amino acids 1-93 and 360-393, respectively) are the important regions that mediate droplet formation. Phase separation was also shown to be regulated by molecular crowding agents, nucleic acids, and posttranslational modification [65]. Interactions with DNA prevent droplet formation [65], but it is still unclear whether RNA molecules have a distinct effect. p53 interacts with DNA via the DBD region, and this interaction is impaired by the R248Q mutation, which significantly increases its amyloidogenic potential [22,50]. Interestingly, this R248Q mutant, but not wt p53, was shown to phase separate in the cytoplasm of HCC70 cells [67]. Therefore, the interaction with polyanions, mainly via the DBD region, as well as changes in protein stability, might modulate the phase separation and phase transition of p53.

It has been reported that RNA can interact and induce p53 aggregation, regulating its DNA-binding activity [68] and, probably, contributing to the oncogenic pathway of p53. Our group showed that, at low RNA:protein ratios, large p53 aggregates are formed in vitro [69]. However, when the RNA concentration is increased, aggregation is reduced [69]. These findings provide insight into the potential interplay between p53 and RNA. The nucleus accommodates a high concentration of nucleic acids including RNA molecules. Thus, high concentrations of RNA might favor the formation of small p53 oligomers rather than higher-order aggregated species. The smaller the species, the higher the chances of p53 being active to bind and regulate gene expression, leading to GoF phenotypes.

Higher temperatures increase p53 transitions with the formation of an anomalous liquid condensate in equilibrium with dispersed p53 molecules [70]. These condensates of p53 were reported to behave as precursors that initiate p53 fibril assembly. Consistently, wt p53 puncta are detected in the cytosol, suggesting the formation of p53 liquid condensates [43]. As DNA and RNA, other polyanions are known to interact with p53. Glycosaminoglycans (GAG) are negatively charged glycans due to the presence of carboxylic and sulfate groups. Chondroitin sulfate A (CSA), a GAG formed by disaccharide units of glucuronic acid and N-acetylgalactosamine with a sulfate at the C4 position, was shown to modulate p53 aggregation [51]. When CSA is mixed with equimolar amounts of the p53 DBD in vitro, it favors a liquid-to-solid phase transition and the formation of amyloid filaments [51]. Notably, CSA is increased in cancer tissues [71], and amyloid filaments were recently extracted from breast and lung carcinoma cells [51], raising the possibility that GAGs might participate in p53 amyloid formation. On the other hand, heparin, a GAG formed mainly by disaccharide units of tri-sulfated iduronic acid and glucosamine, appears to inhibit growth and metastasis in primary tumors and improve patient survival [72,73]. Bemiparin, a low molecular weight heparin with clinical use, induces the expression of different caspases and p53, demonstrating an effect on cell cycle and apoptosis in hepatocellular carcinoma cells [74]. Likewise, berberine nanoparticles loaded with heparin induce osteosarcoma cell apoptosis [75].

Hyaluronic acid (HA) is another type of GAG that is anionic in nature but unsulfated. Extreme high-molecular-weight HA (EHMW-HA) have been connected to an extracellular matrix composition that is characteristic of tumor resistant animals [76]. Extreme high-molecular-weight HA was also shown to induce p53 expression, enhancing breast cancer cell apoptosis [77]. However, it remains to be determined whether the anti-tumor effects observed with GAGs are also related to the modulation of p53 aggregation and its reactivation. To date, a role of GAGs in modulating p53 amyloid conversion has not been confirmed using in cell-based approaches.

## 5. p53 Aggregation Mechanism and Treatment Options

The numerous studies on p53 DBD aggregation suggest that it follows the general model of amyloid formation: monomer → misfolded monomer → dimer → oligomer → protofibril → mature fibril (Figure 4). Under sub-denaturing concentrations of guanidine hydrochloride, the structural ensemble of the p53 DBD can be shifted to populate swollen molten globule states with native like secondary and tertiary structure, and accelerated aggregation kinetics [78]. An open question is to define the oligomeric/multimeric species of mutant p53 or altered p53 that exert GoF activity resulting in higher malignancy. In our recent study, oligomers of the M237I mutant were visualized in the nucleus of living cells [43]. This mutant is associated with aggressive glioblastoma and confers resistance to temozolomide. It is tempting to think that small oligomers are sustained by interaction with nucleic acids, especially RNA, most likely in a phase-separated condition found in membraneless organelles, such as in nuclear bodies and in the nucleolus [62,65]. Whether the phase separation of p53 precedes its aggregation to amyloid oligomers deserves further investigation. Structurally destabilizing p53 DBD mutations can result in a similar outcome [43]. As such, native-state stabilization is a primary avenue being explored to impede p53 DBD aggregation [23,78]. Most approaches to date endeavor to use small molecules to stabilize particular mutants, with the majority of efforts focusing on the cavity induced by the Y220C hotspot mutation [79]. Due to the high number of destabilizing p53 DBD mutations, a more general solution could be advantageous. PRIMA-1 and resveratrol are possible candidate molecules to this end [80,81]. In the case of PRIMA-1, our group evidenced the molecular mechanism through which the compound rescues amyloid aggregates of mutant p53 and thereby decreases the dominant negative (DN) and gain-of-function (GoF) effects [80], indicating that mutant p53 aggregation is an exceptional target for the development of new drugs against cancer.

A better understanding of the molecular basis of p53 DBD instability would likely facilitate the development of broad-spectrum native-state stabilization approaches. Compared to its family members, p63 and p73, the p53 DBD is considerably more susceptible to pressure-induced unfolding and aggregation, which correlates well with its poorer backbone hydrogen bond protection, leading to greater hydration and structural destabilization [82]. Precise mapping of the structural weak spots may be a potential route for developing generalized strategies for stabilizing the p53 DBD. Another direction being explored to impede p53 DBD aggregation is targeting of aggregates. The ReACp53 peptide, which is based on the p53 DBD aggregation sequence, and various small molecules are able to bind aggregates and inhibit their propagation [83,84]. The development of this class of aggregation inhibitors could perhaps benefit from having a better description of p53 DBD aggregate structure. A major challenge in obtaining such detail is structural polymorphism, as even short peptides corresponding to the p53 DBD aggregation sequence present varied topologies and fibril profiles [83,85] (Figure 4). Furthermore, amyloid polymorphism can also be sensitive to environmental factors such as pH, temperature, and ionic strength, which have not been comprehensively studied for p53 aggregates. Moving forward, development of direct aggregation inhibitors may be aided by determination of the physiological conditions that trigger p53 amyloid conversion, and the atomistic structures of the pathological species.

## Figures and Tables

**Figure 1 biomolecules-10-00548-f001:**
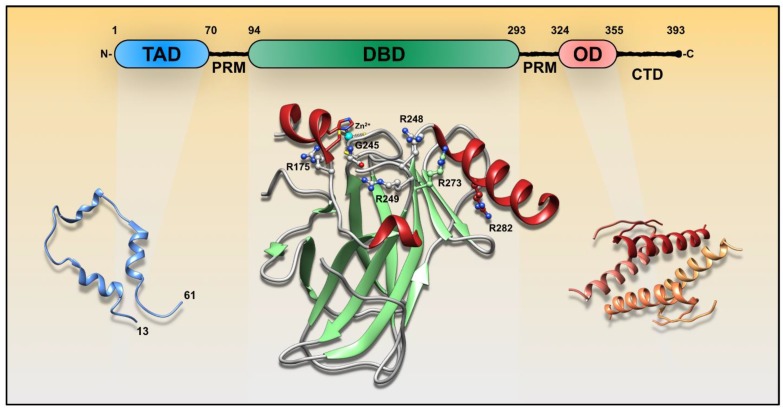
Structure organization of p53 and hotspot mutated sites in cancer. TAD, transactivation domain; PRM, proline-rich motif; DBD, DNA-binding domain; OD, oligomerization domain; CTD, C-terminal domain. (PDB codes: TAD-2L14; DBD-2FEJ; OD-1OLG).

**Figure 2 biomolecules-10-00548-f002:**
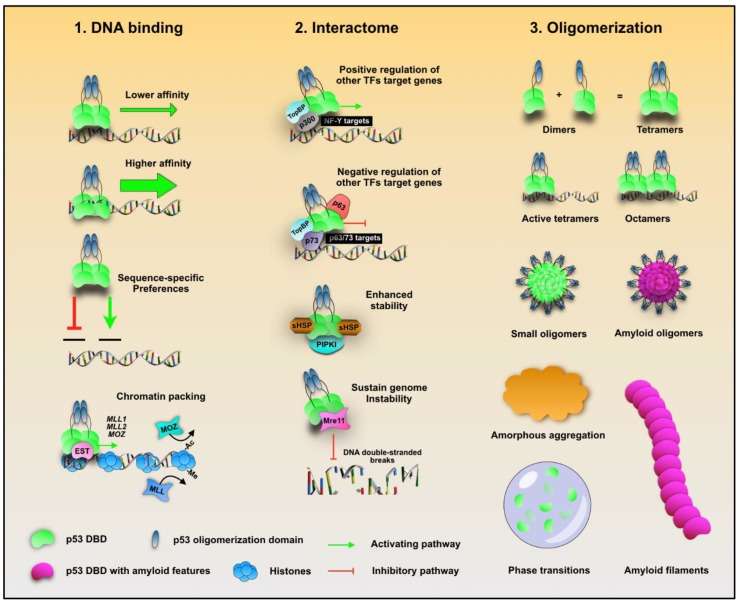
The spectrum of outcomes arising from p53 mutations. Cancer-related mutations affect p53 in multiple ways such as its properties and preferences to bind DNA, its cellular partners, and the propensity to associate itself forming a broad range of oligomeric species and phase transitions. TFs, transcription factor.

**Figure 3 biomolecules-10-00548-f003:**
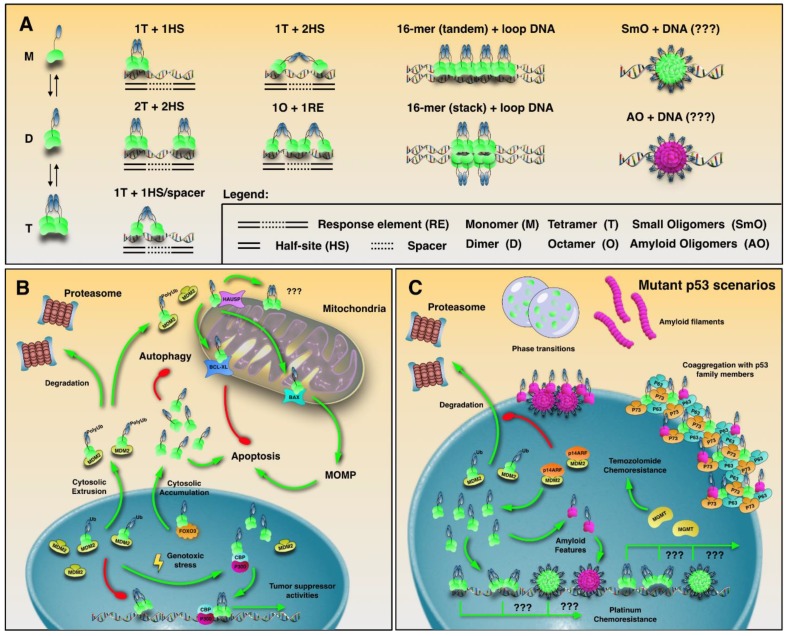
DNA binding and p53 signaling in the context of its oligomeric species in cancer. (**A**) Different p53–DNA interactions reported thus far. (**B**) p53 signaling emphasizing its non-transcriptional activities. (**C**) Mutant p53 signaling within the context of cancer and its aggregates. Question marks (???) show potential events not yet reported by experiments. Green and red traces show activating and inhibitory pathways, respectively.

**Figure 4 biomolecules-10-00548-f004:**
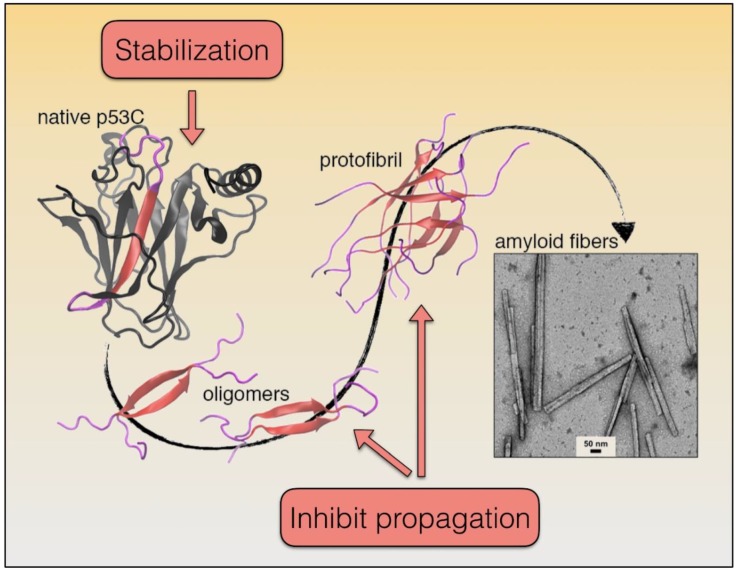
Illustration of p53 aggregation. The scheme shows a potential route of p53 amyloid aggregation and potential avenues of inhibition based upon molecular dynamic simulations and electron microscopy experiments [85].

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
