# Peer review of "The Status of p53 Oligomeric and Aggregation States in Cancer"

_biomolecules, 2020, doi:10.3390/biom10040548_

Round 1

Reviewer 1 Report

Major points include:

  1. The consequences of p53 mutations has been reviewed in a number of papers. This current review aims to summarize the current knowledge regarding p53 aggregation. However, this review does not provide a systemic overview of p53 aggregation and fails to provide the implications of research findings.
  2. The mechanisms underlying p53 aggregation and how aggregation affects the function of p53 are not well discussed. This review lacks a major clue to connect different parts together.
  3. It will be better if the authors could discuss the results from individual studies and how these results help the authors to draw the conclusion. In other words, the authors need to discuss the cited studies in detail and provide explanations about what these results mean.
  4. The manuscript is densely written and is difficult to follow through. As a review article, it will be helpful if the authors could make it clear about what they would like the readers to grasp from this article.

Author Response

Reviewer 1:

  1. The consequences of p53 mutations has been reviewed in a number of papers. This current review aims to summarize the current knowledge regarding p53 aggregation. However, this review does not provide a systemic overview of p53 aggregation and fails to provide the implications of research findings.

Response: Our review aims to summarize the roles of p53 oligomeric states in cancer. We discuss both folded states, such as dimers and tetramers, and aggregates. Extensive changes have been made to the content and organization of the review to improve its readability and highlight the implications of the different works that are touched upon. The revised manuscript now discusses different oligomeric states (dimers, tetramers, aggregates) separately, while outlining the cause, implications, and possible ways of addressing the issue.

  1. The mechanisms underlying p53 aggregation and how aggregation affects the function of p53 are not well discussed. This review lacks a major clue to connect different parts together.
  2. It will be better if the authors could discuss the results from individual studies and how these results help the authors to draw the conclusion. In other words, the authors need to discuss the cited studies in detail and provide explanations about what these results mean.

Response: We have expanded the discussion and interpretation of several of the studies referred to in the review.

Now within page 7 of 21: In cell-based assays, while transiently over expressed wt and p53 DNA-contact mutations revealed monomers, tetramers, and octamers on western-blots, aggregating hotspot mutations including R175H and R249S revealed larger multimeric assemblies. Overall, these demonstrations illustrate that multiple oligomeric states are able to assemble and potentially bind to DNA, regulating gene expression.

Now within page 9 of 21: By means of immunofluorescence, we discovered that the M237I p53 mutation colocalizes with amyloid oligomers in these glioblastoma cells. Further, we observed by fluorescence correlation spectroscopy that this chemoresistant p53 mutation populates oligomers larger than tetramers within the nuclei of living cells. This finding links the role of oligomers as transcriptionally active species. It is possible that some oligomeric or aggregated p53 species act as protagonists to GoF activities, such as chemoresistance. This might occur by direct binding to novel DNA regulatory elements, impacting the expression of genes related to the oncogenic phenotype, binding to other nuclear or cytosolic partners, or by other non-transcriptional activities of aggregated p53 species in the cytosol. These ideas certainly warrant further investigation.

  1. The manuscript is densely written and is difficult to follow through. As a review article, it will be helpful if the authors could make it clear about what they would like the readers to grasp from this article.

Response: The abstract has been modified as follows to clarify the principal message of the review.

Despite being referred to as the guardian of the genome, when impacted by mutations, p53 can lose its protective functions and become a renegade. The malignant transformation of p53 occurs on multiple levels, such as altered DNA binding properties, acquisition of novel cellular partners, or associating into different oligomeric states. The consequences of these transformations can be catastrophic. Ongoing studies have implicated different oligomeric p53 species as having a central role in cancer biology; however the correlation between p53 oligomerization status and oncogenic activities in cancer progression remains an open conundrum. In this review, we summarize the roles of different p53 oligomeric states in cancer, and discuss potential research directions for overcoming aberrant p53 function associated with them. We address how misfolding and prion-like amyloid aggregation of p53 seem to play a crucial role in cancer development. The misfolded and aggregated states of mutant p53 are prospective targets for the development of novel therapeutic strategies against tumoral diseases.

Reviewer 2 Report

Oliveira et al. aim to describe the state of the literature for p53 oligomerization/aggregation.    They present a very nice and quite thorough review which I have only minor critiques and suggestions for. 

Minor comments

Lines 97-99: The authors contrast loss and gain of function mutations.  It is more correct to describe alterations of p53 as causing both loss (of normal) and gain (of abnormal tumor promoting) functions, rather than as pure GoFs.

Lines 132-135:  The authors assert GoF p53 mutations do not abrogate p53 DNA binding.  I believe this is incorrect – many p53 mutants that are described as having gain of function (eg. R248Q/W, R273H/C) do not bind DNA.  They can associate with chromatin through different co-factors, but the protein itself is no longer a sequence specific DNA binding protein.

Lines 183-193: The authors should briefly summarize their data supporting the role of aggregation in cancer. They return to this later, but it would be helpful to give a summary here.

Lines 328-330: The authors propose that small oligomers might be preferentially formed by p53 in nucleic acid dense regions such as the nucleus, and that this might favor a regulatory role.  I am not sure I follow this logic or entirely agree, the authors should further explain this reasoning.

Lines 366-369: The authors briefly mention their own study on M237I, they should either expand this description or remove it.  As currently presented it does not contribute.

Lines 378-380: The authors should note that reactivation (especially with PRIMA) is an area of controversy with many conflicting results.

Author Response

Reviewer 2:

Lines 97-99: The authors contrast loss and gain of function mutations. It is more correct to describe alterations of p53 as causing both loss (of normal) and gain (of abnormal tumor promoting) functions, rather than as pure GoFs.

Response: Text was rewritten to clarify this point.

Now read: “p53 mutations can impact its functionality in different ways. They may lead to a loss-of-function phenotype, where p53’s ability to bind DNA is compromised [27]. Second, they may alter p53’s conformation, leading to gain-of-function (GoF), or oncogenic activity [2].”

Lines 132-135: The authors assert GoF p53 mutations do not abrogate p53 DNA binding.  I believe this is incorrect – many p53 mutants that are described as having gain of function (eg. R248Q/W, R273H/C) do not bind DNA. They can associate with chromatin through different co-factors, but the protein itself is no longer a sequence specific DNA binding protein.

Response: Text was rewritten to clarify this point.

Now read: “P53 mutations do not necessarily abolish its DNA-binding capabilities, but in some GoF p53 mutations, substitution completely abrogates the ability to bind DNA.”

Lines 183-193: The authors should briefly summarize their data supporting the role of aggregation in cancer. They return to this later, but it would be helpful to give a summary here.

Response: The review was modified in several aspects to better support the role of aggregation in cancer. 

Lines 328-330: The authors propose that small oligomers might be preferentially formed by p53 in nucleic acid dense regions such as the nucleus, and that this might favor a regulatory role.  I am not sure I follow this logic or entirely agree, the authors should further explain this reasoning.

Response: Nucleic acids themselves working as solvents in a crowded milieu would limit the formation of larger aggregates. In that sense, small oligomers are more likely species to be formed under such conditions.     

Lines 366-369: The authors briefly mention their own study on M237I, they should either expand this description or remove it. As currently presented it does not contribute.

Response: We expanded several explanations of our study to better contribute to the scope of this review.

Lines 378-380: The authors should note that reactivation (especially with PRIMA) is an area of controversy with many conflicting results.

Response: Although the most effective re-activation strategies are yet unclear, the reactivation of p53 using PRIMA-1 has been studied by various research groups, none of which present conflicting results. In fact, several papers are consistent in showing PRIMA-1 to be an effective compound for reactivation. In their article “PRIMA-1 reactivates mutant p53 by covalent binding to the core domain”, Lambert and Gorzov  show that PRIMA-1 can be transformed into adducts with thiols. This covalent modification re-activated p53. Furthermore, de Costa and Campos provide evidence that p53 can be modulated using resveratrol. These findings in turn improve our understanding of reactivation by exploring aggregation mechanisms.

Round 2

Reviewer 1 Report

The authors have modified the text. The manuscript has been improved.